# Vertical Air Motion Retrievals in Deep Convective Clouds using the ARM Scanning Radar Network in Oklahoma during MC3E

Kirk W. North<sup>1</sup>, Pavlos Kollias<sup>1, 2</sup>, Scott E. Giangrande<sup>3</sup>, Scott M. Collis<sup>4</sup>, and Corey K. Potvin<sup>5</sup>

<sup>1</sup>Department of Atmospheric and Oceanic Sciences, McGill University, Montreal, Quebec

<sup>2</sup>School of Marine and Atmospheric Sciences, Stony Brook University, Stony Brook, New York

<sup>3</sup>Atmospheric Sciences Division, Brookhaven National Laboratory, Upton, New York

<sup>4</sup>Environmental Science Division, Argonne National Laboratory, Lemont, Illinois

<sup>5</sup>Cooperative Institute for Mesoscale Meteorological Studies, and NOAA/OAR/National Severe Storms Laboratory, and School of Meteorology, University of Oklahoma, Norman, Oklahoma

*Correspondence to:* Kirk W. North (kirk.north@mail.mcgill.ca)

**Abstract.** The U.S. Department of Energy (DOE) Atmospheric Radiation Measurement (ARM) program's Southern Great Plains (SGP) site includes a heterogeneous distributed scanning Doppler radar network suitable for collecting coordinated Doppler velocity measurements in deep convective clouds. The surrounding National Weather Service (NWS) Next Generation Weather Surveillance Radar 1988 Doppler (NEXRAD WSR-88D) further supplements this network. Radar velocity

- measurements are assimilated in a three-dimensional variational (3DVAR) algorithm that retrieves horizontal and vertical air motions over a large analysis domain (100 km  $\times$  100 km) at storm-scale resolutions (250 m). For the first time, direct evaluation of retrieved vertical velocities with those from collocated 915-MHz radar wind profilers is performed. Mean absolute and root-mean-square differences between the two methods are on the order of 1 m s<sup>-1</sup> and 2 m s<sup>-1</sup>, respectively. Moderate time-height correlations on the order of 0.5 are also shown to exist between the two methods. An empirical sensitivity analysis
- is done to determine a range of 3DVAR constraint weights that adequately satisfy both velocity observations and anelastic mass continuity. It is shown that the vertical velocity spread over this range is on the order of  $1 \text{ m s}^{-1}$ . A similar sensitivity analysis reveals that iterative multi-Doppler techniques have difficulty satisfying velocity observations and mass continuity simultaneously. These results provide a form of assurance in the use of 3DVAR retrieved vertical velocities for evaluating numerical simulations of deep convective clouds.

# 15 1 Introduction

The representation of deep convection at cloud resolving and global circulation model scales (CRMs and GCMs) remains a serious challenge (Lin et al., 2006; Jakob, 2010). Part of the challenge can be attributed to the lack of comprehensive observations of dynamics and microphysics in these vigorous cloud systems (Ferrier, 1994; Milbrandt and Yau, 2005; Mrowiec et al., 2012). In particular, cloud dynamical insights may provide necessary guidance for improving these simulations to storm scales

and act as a basis for improving convective parameterizations at GCM scales (Lang et al., 2007; Wu et al., 2009; Nicol et al., 2015).

Despite the importance of vertical velocity measurements in deep convection, such measurements are not easy to acquire. Aircraft penetration of convective clouds offer the most direct method to measure these vertical air motions (Lenschow, 1976), however, practical hazards and operational costs have resulted in a valuable but limited dataset (e.g., Byers and Braham, 1948; LeMone and Zipser, 1980; Donner et al., 2001). Profiling Doppler radars have a vertical velocity retrieval uncertainty on the order of  $1-2 \text{ m s}^{-1}$  in convective clouds, thus offer a viable substitute for in situ aircraft measurements (Jorgensen and LeMone,

- 1989; Cifelli and Rutledge, 1994; May and Rajopadhyaya, 1999; Williams, 2012; Heymsfield et al., 2010; Giangrande et al., 2013a; Kumar et al., 2015). Furthermore, profiling radars provide a high degree of detail of convective clouds in both time and height, and can sample even the most intense convective cores. However, profiling radars have a limited use in direct model evaluation due to their narrow view of these large, three-dimensional systems.
- Scanning Doppler radars and the use of multi-Doppler retrieval techniques may help overcome known in situ aircraft and profiling radar sampling limitations. In addition to improved statistics of deep convection, this measurement approach offers the ability to document the three-dimensional structure of updrafts and downdrafts. However, the use of multi-Doppler retrieval techniques is not straightforward. First, the number of radars in the network and their respective locations has a direct impact on the retrieval quality. Second, distributed Doppler radar networks, including mobile radar deployments, are not widely available
- or standardized. Operational radar networks tend to provide inadequate coverage throughout the depth of deep convective clouds, particularly at cloud top, necessary to properly constrain vertical velocity retrievals. Third, several multi-Doppler retrieval techniques have been proposed. The various techniques can be categorized as either iterative or simultaneous based on their treatment of mass continuity.

Iterative techniques solve the integral mass continuity equation throughout the column, contingent on known vertical velocity boundary conditions at the bottom level (e.g., upwards integration) and/or top level (e.g., downwards integration) (e.g., O'Brien,

- 1970; Ray et al., 1980; Protat and Zawadzki, 1999). This first requires an estimate of horizontal wind divergence at each level made in a previous step, hence the non-simultaneity of iterative techniques (Dowell and Shapiro, 2003; Potvin et al., 2012a). By their nature, iterative techniques propagate information in one direction, thus errors in horizontal wind divergence accumulate throughout the column, which in turn leads to larger errors in vertical velocity (e.g., Ray et al., 1980).
- Simultaneous techniques treat mass continuity similar to other analysis constraints by inserting it directly into the cost function. This avoids accumulation of errors throughout the column since mass continuity is analyzed everywhere simultaneously. Moreover, these techniques are known to mitigate retrieval instabilities in poorly constrained regions like the dual-Doppler radar baseline (Bousquet and Chong, 1998; Dowell and Shapiro, 2003). Since simultaneous techniques are by definition 3DVAR techniques, we will refer to them as such throughout the remainder of this study.

20

Several studies have investigated multi-Doppler wind retrieval uncertainties by identifying or utilizing (i) importance of Doppler radar measurement errors and beam geometry (e.g., Doviak et al., 1976; Nelson and Brown, 1987; Matejka and Bartels, 1998) (ii) influence of radar data objective analysis (e.g., Clark et al., 1980; Gal-Chen, 1982; Testud and Chong, 1983; Chong et al., 1983; Given and Ray, 1994; Majcen et al., 2008; Shapiro et al., 2010; Collis et al., 2010) and (iii) observing system simulation experiments (OSSEs) (e.g., Fanyou and Jietai, 1994; Gao et al., 1999; Liou and Chang, 2009; Potvin et al.,

2012b; Potvin and Wicker, 2012). However, few studies have explored practical retrieval performance to other independent air motion estimates from aircraft or ground-based profiling radars (e.g., Collis et al., 2013; Newsom et al., 2014)

While 3DVAR wind retrievals have been studied using OSSEs, an implementation, verification and sensitivity analysis on actual datasets is noticeably missing. The ARM SGP site in Oklahoma provides an excellent opportunity to investigate

- the benefits and relevant issues associated with multi-Doppler wind retrievals (Mather and Voyles, 2012). The SGP site was recently upgraded to include the installation of a distributed scanning Doppler radar network. The wind retrievals presented in this study were derived from scanning radar data collected during the Midlatitude Continental Convective Clouds Experiment (MC3E), a joint field campaign between the DOE ARM program and the National Aeronautics and Space Administration (NASA) Global Precipitation Measurement (GPM) mission Ground Validation (GV) program (Jensen et al., 2015a). This
- campaign also featured the deployment of four radar wind profilers, each operating in a novel deep convective mode (Tridon et al., 2013; Giangrande et al., 2013a). The four radar wind profilers were optimally placed within the scanning radar network to provide independent validation for the multi-Doppler vertical air motion retrievals.

The remainder of this paper is organized as follows. A description of the dataset and radar data processing is presented in Section 2. Section 3 provides the background for 3DVAR wind retrievals from multiple scanning Doppler radars. Retrieval

sensitivity and a method for producing physically sound wind fields is discussed in Section 4. Section 5 presents 3DVAR wind retrieval results in the context of how they compare with those from independent collocated radar wind profilers as well as with an iterative upwards integration method. Section 7 is reserved for summary and concluding remarks.

#### 2 Dataset and radar processing

The MC3E took place during April-June 2011 in northern Oklahoma and surrounding states. A total of five events from MC3E were analyzed for this study and are listed in Table 1. These events represent a variety of warm season convection over Oklahoma, including nocturnal elevated convection (25 April 2011), widespread stratiform precipitation with embedded convection (11 May 2011), mesoscale convective system (MCS) and associated squall line (20 May 2011), and isolated severe supercell thunderstorms (23-24 May 2011). The approximate time frame defining each event reflects profiling radar observations recorded at the SGP Central Facility (CF).

#### 25 2.1 ARM radar network

The ARM SGP site features a network of scanning Doppler and dual-polarization radars capable of providing coordinated coverage of cloud systems over a large domain. The locations of scanning and profiling radars around the SGP CF during MC3E are shown in Fig. 1. The radar facility includes a 6.3-GHz C-band scanning ARM precipitation radar (CSAPR) and three 9.4-GHz X-band scanning ARM precipitation radars (XSAPRs). During the MC3E, a specific deep convection volume

30 coverage pattern (VCP) was implemented in an attempt to provide dense coverage throughout the depth of typical warm season Oklahoma convection. The technical specifications of the ARM scanning radars are listed in Table 2.

Data were calibrated and processed according to several standard methods in the literature, including a conventional dualpolarization reflectivity correction for attenuation in rain appropriate for the ARM C-band radar (Bringi and Chandrasekar, 2001; Giangrande et al., 2013b, 2014; Helmus and Collis, 2016). Aliased radial velocity measurements were corrected using the four-dimensional technique described in James and Houze (2001). Similar to Collis et al. (2013), this technique was applied iteratively using multiple wind profiles from the MC3E radiosonde network to produce robust results (e.g., Jensen et al., 2015b).

Finally, each radar volume was manually inspected to check for conspicuous errors and artifacts.

Following these standard procedures, radar data were mapped to a common Cartesian analysis domain. The domain for this study covers 100 km x 100 km x 10 km in meridional, zonal, and vertical extent, respectively, with 250 m grid spacing in each dimension. The horizontal area covered by the grid approximately encloses all available XSAPR coverage as shown in Fig. 1,

with the SGP CF located at the origin. Since the surface elevation of the analysis domain varies less than 30 m over its entire extent, this study neglects nuances associated with complex terrain (e.g., Chong and Cosma, 2000; Liou et al., 2011). Radar data are mapped using a two-pass isotropic Barnes distance-dependent weight with a constant smoothing parameter  $\kappa = 2 \text{ km}^2$ and convergence parameter  $\gamma = 0.5$  (e.g., Trapp and Doswell, 2000; Majcen et al., 2008),

$$W_{i,q}(d) = \exp\left(\frac{-d_{i,q}^2}{\kappa\gamma^{p-1}}\right) \quad \forall \quad i = 1, \dots, n \text{ and } q = 1, \dots, Q$$

$$\tag{1}$$

Here W<sub>i,q</sub> is the weight for grid point *i* and radar gate *q* separated by distance *d* for pass *p* = 1 or 2. The cutoff distance defining the *Q* closest radar gates is the distance where the weight effectively vanishes, which is *d* ≈ 4 km. Several choices for weighting functions and their free parameters are found throughout the literature (e.g., Cressman, 1959; Barnes, 1964; Pauley and Wu, 1990; Askelson et al., 2000, 2005; Askelson and Straka, 2005; Trapp and Doswell, 2000), however the weighting function used in this study is desirable for the preservation of phase and amplitude information of the input radar data, as well
as its relative insensitivity to the spatial characteristics of the input data (Trapp and Doswell, 2000).

Four 915-MHz UHF-band ARM zenith-pointing radar wind profilers (UAZRs) were placed within the scanning radar network and their locations are also shown in Fig. 1. The technical details of the radar wind profilers running in a novel convection mode during MC3E are also given in Table 2. Vertical air motion retrievals from the wind profilers follow the method outlined by Giangrande et al. (2013a) and are assumed accurate to within  $1-2 \text{ m s}^{-1}$  in deep convective drafts. An example of UAZR observations of convective clouds observed on 25 Apr 2011 and the corresponding vertical air motion retrieval are shown in

observations of convective clouds observed on 25 Apr 2011 and the corresponding vertical air motion retrieval are shown in Fig. 2. The scales of motion resolved by the UAZRs are inherently different than that of the scanning radars, requiring the UAZR data to be filtered in time-height before any comparison with scanning radar retrievals can be made (see Section 5).

#### 2.2 NEXRAD WSR-88D network

The NEXRAD WSR-88D S-band radar network surrounding the SGP site provides additional coverage and robust reflectivity measurements for each event listed in Table 1. This is especially true for 11 May 2011, where CSAPR-I7 was nonoperational yet WSR-88D reflectivity measurements were available and less susceptible to attenuation in rain than the XSAPRs.

5

20

The closest WSR-88D site to the SGP CF is Vance Air Force Base (KVNX), located approximately 56 km west of the CF (see Figure 1). This relatively large distance, coupled with the 0.5 deg base elevation scan of KVNX, ensures that its transmitted pulses are already 1 km above the surface directly over the CF. Figure 3 shows the nearest neighbour distance between radar gates and grid points for the 20 km x 20 km surrounding the CF assuming standard atmospheric refraction and Earth curvature (e.g., Doviak and Zrnić, 1993). The circular features seen in most cross sections are a result of discrete elevation scans. Between the surface and 2 km AGL the ARM radars have enhanced coverage compared to KVNX. In particular, the ARM radars provide coverage that is ideal for characterizing the planetary boundary layer (PBL) since nearest neighbours are typically less than 150 m away from each other within this layer. At heights above approximately 2 km AGL, KVNX becomes increasingly valuable, especially for grid points close to and directly above the ARM radars. The dark red shades ( $d \ge 2$  km)

10 seen in the XSAPR panels of Fig. 3 highlight the radar cone of silence, a measurement gap due to no elevation scans past 50 deg (see Table 2).

#### 3 3DVAR wind retrieval methodology

Results presented in this study capitalize on the physical constraints of radial velocity observations, anelastic mass continuity, surface impermeability, background wind field, and spatial continuity. Assuming  $\boldsymbol{u} = [u_1, \dots, u_n]$ ,  $\boldsymbol{v} = [v_1, \dots, v_n]$ , and  $\boldsymbol{w} =$ 15  $[w_1, \dots, w_n]$  are the eastward, northward, and vertical wind components respectively, we have the cost function

$$J(\boldsymbol{u}, \boldsymbol{v}, \boldsymbol{w}) = J_o + J_c + J_p + J_b + J_s \tag{2}$$

The optimal wind field solution is at the (global) minimum of J which implies the gradient of J with respect to u, v, and w vanishes. For applications requiring large-scale (e.g.,  $1 \times 10^6$  variables) nonlinear cost functional minimization, it is often necessary to use an iterative conjugate-gradient algorithm (Navon and Legler, 1987). In Gao et al. (1999), where a similar cost function and conjugate-gradient minimization algorithm were used, u and v were found to be well recovered within the first 50 minimization iterations, however w lacked both coherency and strength until 200+ iterations. We use these values as a reference point for the minimum number of iterations required to minimize Eq. (2).

#### 3.1 Radial velocity observations: $J_o$

With radial velocity observations  $\tilde{v_r}$  defined on the same *n*-point grid there is no need for an observation operator found in 25 general 3DVAR schemes, and the observation constraint in Eq. (2) is instead given by

$$J_o = \frac{1}{2} \sum_{l=1}^{m} \left[ \left( \boldsymbol{v}_r - \tilde{\boldsymbol{v}}_r \right)^T \boldsymbol{\Lambda}_o \left( \boldsymbol{v}_r - \tilde{\boldsymbol{v}}_r \right) \right]$$
(3)

The sum is over the *m* radars used in the retrieval.  $\Lambda_{o}$  is an  $n \times n$  matrix analogous to the inverse observation error covariance matrix in general 3DVAR schemes with two assumptions. The first is observation errors are uncorrelated meaning  $\Lambda_{o}$  is a

diagonal matrix (this assumption extends to all other constraints as well). The second is that radial velocity observations are weighted according to maximum value of Eq. (1) during first pass. This naturally gives more weight to CSAPR and XSAPR observations within the PBL, and effectively ignores mapped observations propagated into sampling gaps such as the cone of silence. Elements of retrieved radial velocity  $v_r$  are

5 
$$v_{r_i} = (u_i \sin \phi_i + v_i \cos \phi_i) \cos \theta_i + (w_i - w_{t_i}) \sin \theta_i \quad \forall \ i = 1, \dots, n$$

$$\tag{4}$$

where  $\phi$  and  $\theta$  are radar azimuth and elevation pointing directions, respectively, and  $w_t$  is the bulk hydrometeor fall speed parameterized using radar reflectivity, temperature, and air density (Caya, 2001). Retrieved wind fields have been shown to be quite insensitive to the choice of fall speed parameterization (e.g., Potvin et al., 2012b).

Radial velocity observations collected from two or more radars sampling the same convective cloud system are used in
Eq. (3). The radial velocity observations are assumed to be closely matched in time. We required that (a) both KVNX and CSAPR-I7 be available (except 11 May 2011) and initiate a volume scan two minutes or less apart, and (b) any complementary XSAPR input initiate from a volume scan two minutes or less from either KVNX or CSAPR-I7. These criteria ignore the issues associated with the advection and evolution of the cloud system (e.g., Gal-Chen, 1982; Shapiro et al., 2009).

#### 3.2 Anelastic mass continuity: $J_c$

15 Anelastic mass continuity is known to be an adequate assumption in deep moist convection (e.g., Ogura and Phillips, 1962; Lipps, 1990). The general form of the mass continuity constraint is given by

$$J_c = \frac{1}{2} L^2 \boldsymbol{D}^T \boldsymbol{\Lambda}_{\mathbf{c}} \boldsymbol{D}$$
<sup>(5)</sup>

where elements of D are the anelastic mass continuity term,

$$D_{i} = \frac{w_{i}}{\rho_{i}} \frac{\partial \rho_{i}}{\partial z} + \frac{\partial u_{i}}{\partial x} + \frac{\partial v_{i}}{\partial y} + \frac{\partial w_{i}}{\partial z} \quad \forall \ i = 1, \dots, n$$
(6)

and all vanish if anelastic mass continuity is perfectly satisfied. L is a length scale inserted to unify the dimensions and magnitude of  $J_c$  with  $J_o$  (e.g., Legler and Navon, 1991; Bousquet and Chong, 1998; Shapiro et al., 2009). For this study we set L = 250 m, which is the grid spacing. In Eq. (6)  $\rho$  is air density derived from the MC3E radiosonde profiles.

Note that for iterative techniques, the vertical extent of the analysis domain controls the possible integration directions. If cloud tops are not adequately contained within the domain, a top boundary condition becomes impossible to define, making downwards integration impractical. For warm season convective clouds in Oklahoma, a domain extending upwards of 15 km AGL may be necessary in order to use downwards integration, however these heights are poorly sampled by the scanning radar network and therefore poorly constrained by observations (see Fig. 3). Furthermore, (Collis et al., 2010) showed that radar mapping artifacts aloft where radar coverage is poor leads to minimum vertical velocity errors on the order of 2 m s<sup>-1</sup> at these heights. This is the primary reason for capping our analysis domain at 10 km AGL.

# 3.3 Surface impermeability: $J_p$

Surface impermeability dictates that w must vanish at the surface so we write

$$J_p = \frac{1}{2} \boldsymbol{w}^T \boldsymbol{\Lambda}_{\mathbf{p}} \boldsymbol{w}$$
(7)

It is treated as a pseudo strong constraint by heavily weighting its impact on surface grid points; non-surface grid points should 5 not be influenced and their weights in  $\Lambda_p$  are set to zero.

#### **3.4** Background wind field: $J_b$

Including a background constraint helps promote a wind field solution in data-sparse regions based on additional observations. The background horizontal wind components  $u_b$  and  $v_b$  are typically those from radiosonde profiles. Since vertical velocity information is unavailable from these sensors, the background constraint is written as

$$J_b = \frac{1}{2} \left[ (\boldsymbol{u} - \boldsymbol{u}_b)^T \boldsymbol{\Lambda}_{\mathbf{b}} (\boldsymbol{u} - \boldsymbol{u}_b) + (\boldsymbol{v} - \boldsymbol{v}_b)^T \boldsymbol{\Lambda}_{\mathbf{b}} (\boldsymbol{v} - \boldsymbol{v}_b) \right]$$
(8)

Since  $u_b$  and  $v_b$  are assumed to be free of systematic errors, they are given the same weight  $\Lambda_b$ .

#### 3.5 Spatial continuity: $J_s$

The spatial continuity constraint is essentially a low-pass filter designed to dampen high frequency perturbations in the wind retrieval. Similar to Gao et al. (1999), we define this constraint as,

$$J_{s} = \frac{1}{2}L^{4}\left[\left(\nabla^{2}\boldsymbol{u}\right)^{T}\boldsymbol{\Lambda}_{su}\nabla^{2}\boldsymbol{u} + \left(\nabla^{2}\boldsymbol{v}\right)^{T}\boldsymbol{\Lambda}_{sv}\nabla^{2}\boldsymbol{v} + \left(\nabla^{2}\boldsymbol{w}\right)^{T}\boldsymbol{\Lambda}_{sw}\nabla^{2}\boldsymbol{w}\right]$$
(9)

In addition to reducing *noise*, Eq. (9) is able to extrapolate a wind field solution into data-sparse or poorly constrained regions. For instance, it may encourage usable solutions along the dual-Doppler baseline or add retrieval value to regions in close proximity to or directly above a radar (Bousquet and Chong, 1998).

## 4 Empirical wind retrieval sensitivity analysis

Typically the constraint weight matrices found in Eq. (3)-(9) are treated as adjustable parameters, controlling the degree to which each constraint influences the final solution. In essence, the values prescribed to each weight are often determined through trial and error (e.g., Gao et al., 1999). Fundamentally, there exists a range of values for each weight that produces a physically sound wind field. A thorough sensitivity analysis could be used to determine this parameter space, but this is often ignored because studies typically consider theoretical wind retrieval performance by comparing it to a known *truth* field

(e.g., model output in an OSSE). The weights that minimize the residual error between the retrieved and truth wind fields are then adopted (e.g., Gao et al., 1999; Potvin et al., 2012a). For applications involving real radar datasets where no truth field is available, one must consider (i) determining the parameter space which produces physically sound wind fields (ii) characterizing the solution spread within the parameter space determined by (i).

5 This section addresses these two points through an extensive sensitivity analysis within the experimental domain indicated by the dashed blue box in Figure 1. This domain has the same 250 m grid spacing and vertical extent as the larger domain, but covers a smaller horizontal area of 20 km x 20 km. Utilizing a smaller domain for the sensitivity analysis reduces processing time and allows for the isolation of specific cloud type regimes (e.g., convective versus stratiform). Since convective air motion retrievals are the primary interest of this study, the sensitivity analysis was done during a time when intense convection filled 10 the experimental domain on 23 May 2011, using scanning radar observations valid between 2236-2243 UTC.

Point (i) is addressed by answering the following two questions. The first is *how well does the wind retrieval satisfy radial velocity observations?* The second is *how well does the wind retrieval satisfy anelastic mass continuity?* The second question is particularly important in the context of numerical modeling and convective parameterizations.

The wind retrieval is said to satisfy the radial velocity observations of one or more radars if the root-mean-square difference (RMSD) between the two is within the uncertainty estimate of the observations themselves. Since it is impractical to account for all sources of error inherent in mapped radial velocity observations, we establish a range of uncertainty and require the RMSD to be within this range. Radial velocity measurement error in regions of low signal-to-noise ratio and large Doppler spectrum width can be as high as  $0.5 \text{ m s}^{-1}$ . The additional uncertainty introduced when mapping irregular radial velocity data to a regular grid is estimated to be on the order of  $1 \text{ m s}^{-1}$ . Therefore, we consider the wind field to satisfy radial velocity observations if it produces a RMSD with one or more radars within 0.5-1.5 m s<sup>-1</sup>, computed over the entire analysis domain.

To determine the degree to which the wind field satisfies anelastic mass continuity, we define the normalized mass continuity residual as

$$\mathbf{NMCR}_{i} = D_{i}^{2} \left[ \left( \frac{w_{i}}{\rho_{i}} \frac{\partial \rho_{i}}{\partial z} \right)^{2} + \left( \frac{\partial u_{i}}{\partial x} \right)^{2} + \left( \frac{\partial v_{i}}{\partial y} \right)^{2} + \left( \frac{\partial w_{i}}{\partial z} \right)^{2} \right]^{-1} \quad \forall \ i = 1, \dots, n$$

$$(10)$$

where  $D_i$  is given by Eq. (6). As NMCR approaches zero, anelastic mass continuity becomes perfectly satisfied. However, this is not necessarily desirable since it is still an underlying assumption. Therefore, we propose a range for NMCR, averaged over the entire analysis domain, between 1-10%, whereby anelastic mass continuity is said to be adequately satisfied.

The response of CSAPR-I7 ṽ<sub>r</sub> RMSD and NMCR to perturbing multiple constraint weights is analyzed, the results of which are shown in Fig. 4. The sensitivity of these two metrics to continuity-background weights is shown in Fig. 4a-b. What is immediately evident in Fig. 4a is the strong dependence of ṽ<sub>r</sub> RMSD on Λ<sub>b</sub>, with little to no dependence on Λ<sub>c</sub>. Even
30 with only a factor of two increase in Λ<sub>b</sub>, the wind retrieval diverges substantially from the radial velocity observations and converges towards the background wind field. As Λ<sub>b</sub> → 0.5, CSAPR-I7 ṽ<sub>r</sub> RMSD approaches the specified upper limit of 1.5 m s<sup>-1</sup>. This is important to note since proponents of 3DVAR methods have reported the relative insensitivity of retrievals to minor changes (e.g., not orders of magnitude) in constraint weights (e.g., Gao et al., 1999; Potvin et al., 2012a). However,

in Fig. 4b,  $\Lambda_{b}$  has a decreased effect on the degree to which the wind retrieval satisfies mass continuity. As expected, this is primarily controlled by  $\Lambda_{c}$ , not only within the continuity-background parameter space but also in the continuity-smoothness parameter space shown in Fig. 4c-f. NMCR is particularly sensitive to  $\Lambda_{c}$  when  $\Lambda_{c} < 250$ . Outside of this range, NMCR is generally more stable with respect to  $\Lambda_{c}$  and NMCR is typically less than 20%. However, as seen in the right column of Fig. 4, in order to obtain NMCR  $\leq 5\%$ ,  $\Lambda_{c}$  must generally be 500 or larger.

A similar sensitivity analysis was done for an iterative upwards integration technique, the results of which are shown in Fig. 5. Similar to the 3DVAR results in Fig. 4a, CSAPR-I7  $\tilde{v_r}$  RMSD is highly dependent on  $\Lambda_b$  and less so on  $\Lambda_c$ , however for  $\Lambda_c > 5$  there is a sharp increase in  $\tilde{v_r}$  RMSD up to approximately  $3 \text{ m s}^{-1}$ , well outside the  $1.5 \text{ m s}^{-1}$  upper limit. In fact, the parameter space in which  $\tilde{v_r}$  RMSD is below  $1.5 \text{ m s}^{-1}$  is very small, and when looked at together with NMCR, no continuity-

- background parameter space exists in which both metrics are reasonably satisfied for an iterative upwards integration technique. It is worth noting that as  $\Lambda_c$  is increased, NMCR appears to asymptote towards a value between 10-15%. This indicates that even in the parameter space where radial velocity observations are effectively ignored (e.g.,  $\tilde{v_r}$  RMSD greater than 3 m s<sup>-1</sup>), iterative upwards integration techniques still have difficulty properly satisfying mass continuity. Results were also poor for the continuity-smoothness sensitivity analysis, in particular they were more unstable, and therefore they are not shown. Therefore,
- we do not expect iterative upwards integration wind retrievals to necessarily satisfy mass continuity, however, radial velocity observations should be reasonably satisfied.

Unlike the continuity-background sensitivity analysis, both metrics appear highly unstable in certain regions of the continuitysmoothness parameter spaces investigated in Fig. 4c-f. For  $\Lambda_{su}$  and  $\Lambda_{sv}$ , which control the degree of smoothing of the horizontal wind components in Eq. (9), CSAPR-I7  $\tilde{v_r}$  RMSD becomes unstable as these two weights approach values of 400 and

- larger. A similar phenomenon occurs for NMCR in Fig. 4d. These highly unstable regions of the parameter space are likely the result of nonlinear effects introduced by the squared second order partial derivatives defined in  $J_s$  and should be avoided altogether. For values of  $\Lambda_{su}$  and  $\Lambda_{sv}$  below approximately 100, CSAPR-I7  $\tilde{v_r}$  RMSD is within 1.5 m s<sup>-1</sup> and relatively stable. However, the parameter space in which this holds true gradually shrinks as  $\Lambda_c$  increases towards 1000. Mass continuity is also adequately satisfied for  $\Lambda_{su} = \Lambda_{sv} < 100$  and  $\Lambda_c > 250$ , with NMCR typically less than 10%. Results for  $\Lambda_{sw}$  are similar to
- those of  $\Lambda_{su}$  and  $\Lambda_{sv}$  except for one aspect. Since  $\Lambda_{sw}$  controls the degree of smoothing of the vertical wind component in  $J_s$ , it has little influence on CSAPR-I7  $\tilde{v_r}$  RMSD since the vertical wind component is generally not well sampled by scanning radars. This manifests itself in Fig. 4e, which shows CSAPR-I7  $\tilde{v_r}$  RMSD to have much less dependence on  $\Lambda_{sw}$  compared to  $\Lambda_{su}$  and  $\Lambda_{sv}$ . A summary of the findings from this sensitivity analysis is recorded in Table 3.

#### 4.1 Vertical velocity solution spread

Each panel in Fig. 4 contains over 2000 wind field realizations, each of which was concurrently saved. Therefore, we compute the 3DVAR vertical velocity solution spread from these thousands of realizations, allowing us to address point (ii) above. It was found that within the range of constraint weights defined in the analysis column of Table 3, the vertical velocity solution spread was relatively narrow at 1.5 m s<sup>-1</sup>. This provides a form of uncertainty estimate for the 3DVAR wind retrievals presented in