# Peer review of "Vertical Air Motion Retrievals in Deep Convective Clouds using the ARM Scanning Radar Network in Oklahoma during MC3E"

_Atmospheric Measurement Techniques, 2016_

## Referee Comment (RC1) · Anonymous Referee #1 · 26 Sep 2016

Review: Vertical Air Motion Retrievals in Deep Convective Clouds using the ARM Scanning Radar Network in Oklahoma during MC3E

Synopsis:

This paper attempts to highlight the capability of the ARM network to gather and used the wealth of data to attempt to address issues with current radar methodology. Data collected during MC3E is analyzed to show how well the ARM network captured three events. Dual-Doppler 3DVAR technique is used show that the results when compared to vertical profiler data located in the vicinity are similar in magnitude, but for one case discussed upward iterative techniques do not perform as well.

Decision: Accept Major

General Comments:

Overall, the work done in this paper is good, but more discussion and expansions of results is needed to make this paper more robust to justify the conclusions that are trying to be made. Especially in the comparison between traditional dual-Doppler methods and 3DVAR methods. I would like to see more details on the quality control methods used rather than just stating 'standard methods' were used.

Need some expansion on the analysis of results and methodology to help readability. The author used capital lambda ($\Lambda$) in the document but in the figures lower case lambda ($\lambda$) was used. Please fix as it was confusing. Chapters 5 and 6 need to be expanded on in areas detailed below and reworked to address readability and clarity.

Major:

1) 2.2: When using KVNX did you change the objective analysis parameters to deal with the coarser resolution and change in spatial coverage of the data from KVNX. It does not appear you did but I would suggest that maybe it is investigated to not introduce artifacts into the Barnes interpolation.
2) Advection correction is ignored for paper, why? Especially with 20 May where KVNX is used. Advection correction has been shown to help improve dual Doppler retrievals. It should be discussed and determined how dealing with advection changes the results between the two retrieval techniques discussed in this paper.
3) Page 7 line 8: What is the spatial/temporal distance from the radiosonde profiles to the analysis grid. Does using the radiosondes as the background create issues or biases in the background due to time and distance from the grid as well as the evolution of the environment over time. Why not use a derived sounding from model analysis (e.g. RAP) as the background?

4) The results and analysis of the three cases are not identically carried out but they should be. Why is the upward iterative technique only compared to one case and not all three? I think comparisons between all cases should be carried out in a similar manner. As well as, similar panel plots for 20 May that is included with the other two cases discussed. There are no reflectivity plots for 20 May that is similar to the other two cases and there should be. Expansion and reworking of 5.2 and 6 will help improve the strength of the paper.

5) I would suggest doing analysis of the differences between the iterative and 3DVAR techniques over all three cases to show differences between the two techniques over three different regimes (QLCS, elevated front, and supercell environments). This would help show the importance of using the 3DVAR technique.

Minor:

Page 2 line 8: Please expand on reasoning why profiling radars provide high detail in time and height and how without the reader knowing which band the radar is operating in how it can sample the most intense convective cores. Feels like there should be a few citations proving this.

In figure 2: Why are there lines of constant height moving radially away from the WSR-88D nearest neighbor distance plot. It looks different than the other radars. I am not sure why similar heights can occur radially outward.

Page 6 line 22: What is the time difference from the air density calculation and the analysis? For example, the air density within the squall line should be very different from the environment ahead of it. When are those values updated and the sensitivity to those changes applied.

Page 8 line 18: Is there a citation or study that shows the velocity measurement error can be 0.5 m s$^{-1}$. Need to justify using 0.5 m s$^{-1}$ as the top of the error.

Page 9 Line 6: I would switch this paragraph with the one after it to help readability.

Page 9 Line 28: Move the last sentence to the beginning of the sensitivity analysis to give the reader the ability to know where to reference as you describe.

Page 10 Line 18: How the new $R_s$ was determined to be 750 m?

Page 11 Line 20-23: These two sentences seem out of order.

Page 11 Line 25: Move the detail about the time axis for the plots higher up to where the plot is initially discussed hear line 20 so the change from the expected axis is noted earlier.

Page 12 Line 19: How do we know that this upwards motion is associated with the squall line convergence zone when there is not plot to give context that relate the two together.

Chapter 6: How many times was the iterative upward technique iterated over the entire column? What was the method used to determine if/when the iterative technique converged on a solution? You mention number of iterations for the 3DVAR technique but none for the upward iterative technique.

Page 14 Line 8-9: This sentence should be removed or changed to not be as strongly worded as there has been other studies where multiband radars have been used together for analysis. See papers resulting from the VORTEX2 field project.

Page 14 Line 27: It is hard for me to accept that the iterative wind retrieval is inferior to the 3DVAR technique from only 1 example and one that did not use advection correction.

---

## Referee Comment (RC2) · Anonymous Referee #2 · 28 Sep 2016

The study investigates high resolution weather radar Doppler observations to estimate vertical air motion in deep convective clouds. Firstly, an empirical wind retrieval sensitivity analysis is presented. Then, the authors evaluate uncertainties and agreement comparing 3DVAR vertical velocity retrievals by X-, C-, and S-band scanning radars with a co-located radar wind profiler. Finally, the authors investigated differences between 3DVAR and an iterative upwards integration retrieval for the squall line event on 20 May 2011.

The study is scientifically interesting, clearly presented by proper language. In the following some remarks are listed to improve the paper.

1)In the conclusions the authors assert "X-, C-, and S-band scanning radars have been

used together to pseudo simultaneously": the reviewer is not able to find information about the S-band radar in Table 2 and its role in the study is not clear. 2) There are no information about radiosonde location, please provide them updating also Figure 1 3) The observation simultaneity is the key factor for multi-Doppler retrievals: please provide more details on this topic. 4) The comparison between 3DVAR and iterative retrievals is performed for only one event. It should extended to all cases.
* * *

---

## Referee Comment (RC3) · Anonymous Referee #3 · 2 Oct 2016

This manuscript does a nice job of describing how observations from several different scanning radar to estimate vertical air motions. I feel the manuscript needs a couple clarifications before being ready to publish.

Specific comments:

1. Page 4, lines 1-7. The first line of page 4 states, "Data were calibrated and processed according to several standard methods in the literature, ...." The manuscripts needs to clarify what ARM data are used in this study. Specifically, were the data calibrated and processed by ARM or by the authors of this manuscript. Also, the manuscript should include the DOI numbers of the ARM datasets used in this study and whether or not the data processed by this research team are available for others to

analyze. Line 7 on page 4 states that the radar data are mapped to a common Cartesian analysis domain. I thought ARM already produced a radial moments to Cartesian coordinate moments data product. The manuscript needs to clarify whether or not this common Cartesian data is the same or different than an ARM data product.

2. Page 4, lines 15 to 19. The manuscript should state the weights used in this study. As written, the manuscript describes that the weights are important, but not the actual weight values. If this work is to be repeatable by others, then the weights of the gridding should be published in this manuscript.

3. Page 4, lines 1 to 19. The manuscript needs to describe in this section what radar observations are used in this manuscript. With the importance of preserving the phase and amplitude information of the input radar data (see line 19), the reader is led to believe that a phase measurement (e.g., Kdp) is used in this study. But in later pages, it appears that only reflectivity and radial velocity are used in this study.

4. Page 5, line 14. This reviewer noticed the phrase "surface impermeability". That is a fancy way of saying "surface boundary condition".

5. Page 5, lines 13-22. I found this section hard to read because the cost function terms are not defined. The manuscript needs to define the cost functions of J0, Jc, Jp, Jb, and Js. As written, these terms are not introduced until subsequent section headings.

6. Page 8, line 25 and onwards. The variable names for the weights are different in the text and in the figures. These different variable names is very confusing for the reader and the manuscript needs to be corrected.

7. Table 2. I found the entries for pulse width and range resolution to be confusing and possibly redundant. The entry for the UAZR range resolution is either 200 meters or 120 meters, not both, please correct or clarify.

---

## Referee Comment (RC4) · Anonymous Referee #4 · 14 Oct 2016

The paper has a thorough introduction explaining the importance and current understanding and well presented graphics. The paper answers many questions I initially would ask – e.g. the attenuation issues and need for S-band in these cases. I appreciated the descriptions of the cases in section 5.2. However I feel there are some "holes" that need to be filled for publication. - Page 4 – the calibration and processing is essentially stated as standard, yet these practices are certainly far from universal and require more detail on what calibration, processing and quality control has occurred. - Equation 1 requires weights, which are subsequently mentioned, but the chosen values are never given – for repeatability they are important. - I found section 3 hard to follow, some symbols seem unspecified (at least until later) or inconsistent (notably

don't match the plots) which is I believe the source of my initial difficulty. - the beginning of section 7 suggests "X-, C-, and S-band scanning radars have been used together to pseudo simultaneously", yet the S-band seems only used for reflectivity in the case where attenuation is thought to be significant, and it isn't clear if it is used at all in other cases. - I'd like to see the S-band radar in table 2 if it is intended to be used together as in the previous point.

---

## Author Comment (AC1) · 18 Jan 2017

The study investigates high resolution weather radar Doppler observations to estimate vertical air motion in deep convective clouds. Firstly, an empirical wind retrieval sensitivity analysis is presented. Then, the authors evaluate uncertainties and agreement comparing 3DVAR vertical velocity retrievals by X-, C-, and S-band scanning radars with a co-located radar wind profiler. Finally, the authors investigated differences between 3DVAR and an iterative upwards integration retrieval for the squall line event on 20 May 2011.

The study is scientifically interesting, clearly presented by proper language. In the following some remarks are listed to improve the paper.

1) In the conclusions the authors assert "X-, C-, and S-band scanning radars have been used together to pseudo simultaneously": the reviewer is not able to find information about the S-band radar in Table 2 and its role in the study is not clear.
2) There are no information about radiosonde location, please provide them updating also Figure 1
3) The observation simultaneity is the key factor for multi-Doppler retrievals: please provide more details on this topic.
4) The comparison between 3DVAR and iterative retrievals is performed for only one event. It should extended to all cases.

Thank you for your comments regarding the quality of our manuscript. Below please find our responses to your remarks.

1) We have included the operational parameters of the S-bands in Table 2. We originally left it out because this radar network is somewhat of a legacy network, i.e., it has been in operation for several decades now, and therefore the information was considered redundant. We have also tried to make it more clear in Section 2.2 that both reflectivity and velocity measurements from the S-band radars are used in our retrievals.
2) Figure 1 has been updated to include radiosonde launch sites during MC3E.
3)  Pg6 Ln9-13 have been edited to include more information about scan time discrepancies between the radars. We are including a new figure which attempts to show this information qualitatively, e.g., average time offset between CSAPR and KVNX as a function of height.
4) We have extended our comparison between 3DVAR and the iterative upwards integration method to all cases.

---

## Author Comment (AC2) · 4 Jun 2017

**Response to Reviewer Comment #1**

We thank this reviewer for their valuable suggestions that have helped to improve this manuscript. Taking into account the comments from all reviewers below, we reorganized several sections of our manuscript. The following document contains our detailed responses to your comments, with our responses in plain text given underneath your original comments in bold type.

**Synopsis:**
**This paper attempts to highlight the capability of the ARM network to gather and used the wealth of data to attempt to address issues with current radar methodology. Data collected during MC3E is analyzed to show how well the ARM network captured three events. Dual-Doppler 3DVAR technique is used show that the results when compared to vertical profiler data located in the vicinity are similar in magnitude, but for one case discussed upward iterative techniques do not perform as well.**

Thank you for your appreciation of our manuscript. Taking into account your suggestions and other reviewer comments, we extended our retrieval analysis to include five events. Those events are listed in Table 1, and new analyses are provided throughout the reorganized sections of our manuscript. Additional details on this change are also shown in responses to reviewer comments below.

**General Comments:**
   **Overall, the work done in this paper is good, but more discussion and expansions of results is needed to make this paper more robust to justify the conclusions that are trying to be made. Especially in the comparison between traditional dual-Doppler methods and 3DVAR methods. I would like to see more details on the quality control methods used rather than just stating 'standard methods' were used.**
   **Need some expansion on the analysis of results and methodology to help readability. The author used capital lambda in the document but in the figures lower case lambda was used. Please fix as it was confusing. Chapters 5 and 6 need to be expanded on in areas detailed below and reworked to address readability and clarity.**

Agree. We extended the comparison (analysis dataset) to the five MC3E cases wherein our datasets were appropriate (e.g., solid collection during deeper convective events).
- The RMSE of radial velocity and the normalized mass continuity residual (NMCR) estimated from the 3DVAR and the iterative upward integration method are now listed in our revised manuscript Table 4.
- A new Figure 6 shows an example of the comparison of retrieved vertical velocity and horizontal divergence, and vertical profiles of RMSE and NMCR for the May 20, 2011 case. The results of our analyses are described in our revised section 5. This portion of our study has been moved from section 6 in the previous manuscript.

We also extended our comparisons of the radar reflectivity and vertical air velocity retrievals between the 3DVAR method and the radar wind profiler retrievals to include all

five MC3E cases.

- The MBD, MAD, RMSD, Spearman's rank correlation ($\rho$), and the Pearson product-moment correlation (r) are listed in revised Tables 5 and 6.
- Figures 7, 9, and 10 show examples of direct comparisons of the time series of reflectivity and vertical velocity retrieved from the wind profiler retrievals and the 3DVAR retrievals for the April 25 and May 20 cases. The comparisons are discussed in section 6 in the revised manuscript.

We also have added improved descriptions about radar data correction / quality control and other factors (in response to other reviewer comments as well) to our revised section 2.

- Radar reflectivity observed by CSAPR-I7 was corrected for attenuation in rain using the CSAPR-I7 specific differential phase (Kdp) measurements as from open-source ARM coding options (e.g., Bringi and Chandrasekar, 2001; Giangrande et al., 2013b, 2014).
- Because XSAPR reflectivities were significantly attenuated in rain and occasionally extinguished to within/behind heavier rain regions, only mean Doppler velocity measurements (not impacted by partial attenuation) were used (as available) in our velocity retrievals. Aliased radial velocity measurements from all radars were corrected / dealiased using the four-dimensional technique described in James and Houze (2001). Each radar volume was manually inspected to check for conspicuous errors and artifacts.

In the manuscript, capital lambda ($\Lambda$) represents a $n \times n$ matrix of constraint weights where $n$ is identical to the number of analysis points, while lower case lambda represents diagonal element of the matrix $\Lambda$, which is treated as adjustable parameters. Although the matrix $\Lambda$ has a number of elements, this study uses constant values of $\lambda$ for $\Lambda_c$, $\Lambda_p$, $\Lambda_b$, and $\Lambda_s$. For $\Lambda_o$, $\lambda_o$ weights are calculated from the nearest neighbor weight and observational data quality based on normalized coherent power for each radar. We specified this in section 3 and revised text, tables, and figures appropriately.

**Major:**
**1) 2.2: When using KVNX did you change the objective analysis parameters to deal with the coarser resolution and change in spatial coverage of the data from KVNX. It does not appear you did but I would suggest that maybe it is investigated to not introduce artifacts into the Barnes interpolation.**

We agree with the reviewer's question. Interpolation methods could produce artifacts such as ring-like structure as pointed out by Collis at al. (2010). The objective parameters should be appropriately set to reduce such artifacts. Collis at al. (2010) in particular presented ring-like artifacts due to data sparsity that appeared above 11 km in altitude. For the present study, we used a constant value for the smoothing parameter $\kappa = 2$ km$^2$ in Eq. (1) for all radars. This value is enough large to collect data in sparse data regions below 10 km altitude (maximum height of the analysis domain in this study) and reduces the ring-like artifact. In the dense data regions, we limited the maximum number of samples for interpolation to nearest 200 to avoid over smoothing. This number 200 could be too large, but we found it

reasonable to allow for a similar behavior over the entire domain. We believe that a more detailed analysis on the effect of changing these smoothing parameters (relative to each radar) on the retrieval results (e.g., an additional sensitivity analysis set) would be better suited as topics for another manuscript.

Moreover, in the 3DVAR algorithm, observation constraint weights are calculated from the nearest neighbor weight and observational data quality based on normalized coherent power for each radar. The nearest neighbor weight at each grid point is calculated by Eq. (1) for the nearest radar data point to the grid point. This can reduce impacts of errors attributable to low radar data sampling rate on the retrieval. We added figures of the nearest neighbor distances and nearest neighbor weights in Figure 3.

**2) Advection correction is ignored for paper, why? Especially with 20 May where KVNX is used. Advection correction has been shown to help improve dual Doppler retrievals. It should be discussed and determined how dealing with advection changes the results between the two retrieval techniques discussed in this paper.**

This is an important question, since we understand that advection could impact our retrieval results and interpretation as compared with other methodologies. We also recognize that it is challenging, but important to correct for advection effects in the multi-Doppler wind retrieval and similarly demonstrate this correction as also meaningful/important as compared to other observational references. While we have attempted to improve our discussion on these ideas, we believe a more complete analysis of these ideas is something suited for a follow-up manuscript. As such, we have decided to present our results and associated discussions of these multi-Doppler radar wind retrievals without including advection correction in this paper (noting that our responses to follow should accompany this manuscript online).

Moreover, since we do not want to pass the buck on this responsibility for this response to reviewers, we have attempted a simple advection correction procedure in which the radar reflectivity and radial velocity were horizontally shifted according to PPI scan time and horizontal winds as observed by the radiosonde. The figure we provide below (Figure R1) shows vertical profiles of NMCR and radial velocity RMSE for the 4 radars we were using for the intense convection case at 1037 UTC on May 20, 2011.

As from the image, the retrieval with advection correction improves the mass continuity residual. However, the advection corrected retrieval produced larger radial velocity RMSE values, and these converged on a larger cost function value (not shown). We suggest this highlights that an advanced topic that cannot be simply addressed as an add-on to the current manuscript. In that sense, our future plans include investigating how the PPI volume scan can reconstruct 3D structure of clouds, taking account of advection and time evolution of clouds using radar forward simulator and high-resolution (0.5 km) and high-frequent (every 20 seconds) model output.

[Figure]

Figure R1: Vertical profiles of NMCR (left) and radial velocity RMSE (right) from the 3DVAR retrievals without advection correction (black lines) and with a simple advection correction (red lines) for 1037 UTC on May 20, 2011.

**3) Page 7 line 8: What is the spatial/temporal distance from the radiosonde profiles to the analysis grid. Does using the radiosondes as the background create issues or biases in the background due to time and distance from the grid as well as the evolution of the environment over time. Why not use a derived sounding from model analysis (e.g. RAP) as the background?**

We thank the reviewer for drawing our attention to this aspect of our manuscript that was not clear. For this effort, we used the ARM-provided Merged Sounding value-added product. This provides a time series of atmospheric moisture, temperature, pressure, and horizontal wind profiles at the SGP CF at 1-minute intervals 266 altitude levels; This product is obtained by using a combination of observations from radiosonde soundings at SGP CF (available every 3 hours during MC3E), microwave radiometers, surface meteorological instruments, and European Centre for Medium Range Weather Forecasts (ECMWF) model output. The product employs a double-sigmoid function for blending weights when deciding between radiosonde and ECMWF profiles (e.g., originated as a way to use the ECMWF to 'gap fill' for times and locations when timely radiosondes were unavailable). The retrieval in this study used a profile nearest the analysis time from the product. We described this in better detail in revised section 2.

**4) The results and analysis of the three cases are not identically carried out but they should be. Why is the upward iterative technique only compared to one case and not all three? I think comparisons between all cases should be carried out in a similar manner. As well as, similar panel plots for 20 May that is included with the other two cases discussed. There are no reflectivity plots for 20 May that is similar to the other two cases and there should be. Expansion and reworking of 5.2 and 6 will help**

**improve the strength of the paper.**

As mentioned in our response to the reviewer's general comment, we agree and have extended the comparison analysis between 3DVAR and upward integration techniques to the five MC3E cases. The RMSD of radial velocity and the normalized mass continuity residual (NMCR) estimated from the 3DVAR and an iterative upward integration method are now provided. Figure 6 shows an example of comparison of retrieved vertical velocity and horizontal divergence, and vertical profiles of RMSE and NMCR for the May 20 case. The result of analysis is described in section 5, which has been moved from section 6 in the previous manuscript.

**5) I would suggest doing analysis of the differences between the iterative and 3DVAR techniques over all three cases to show differences between the two techniques over three different regimes (QLCS, elevated front, and supercell environments). This would help show the importance of using the 3DVAR technique.**

Thank you for the suggestion. As shown in Table 4, the 3DVAR technique provides lower NMCR and radial velocity RMSE values than the upward integration technique for the five cases – but, we would confirm that the NMCR values from the April 25 and May 11 cases are very low, and we find similar retrievals (behaviors) between the two compared retrieval techniques. Note that these two cases are classified as nocturnal elevated convection (April 25) and widespread stratiform precipitation with embedded convection (May 11), respectively. In other words, both cases included arguably weaker convective cell regions, and the propagation speeds for these events were generally also much slower. The remaining cases are of the more typical severe convective and MCS type events, e.g., those having larger areas of stronger, faster moving convective cells. The result may suggest that the iteration upward integration technique is still highly usable and comparable to the 3DVAR approach for the two weaker convective, slowing moving events where the mass continuity equation is a dominant parameter; For the severe convection and MCS cases, the 3DVAR technique is arguably more viable. We have added an improved discussion on these ideas to section 5.

**Minor:**
**Page 2 line 8: Please expand on reasoning why profiling radars provide high detail in time and height and how without the reader knowing which band the radar is operating in how it can sample the most intense convective cores. Feels like there should be a few citations proving this.**

In section 2.4, we have cited Tridon et al. (2013) and Giangrande et al. (2013a), who provided the necessary details on wind profiler operations, configurations and retrieval examples for typical measurements in deep convective cells passing over the SGP Central Facility and extended profiler sites during MC3E. The radar is operating at UHF frequency ranges, which would collect relatively unattenuated (in rain) measurements in deeper convective cells that propagate overhead. Table 2 provides the wind profiler settings for its

'precipitation' radar operation modes, which are significantly different than what is typically assumed as boundary layer wind profile modes of operation.

**In figure 2: Why are there lines of constant height moving radially away from the WSR-88D nearest neighbor distance plot. It looks different than the other radars. I am not sure why similar heights can occur radially outward.**

The figure shows a distance between the grid point and the nearest radar data point at each grid point. In regions where the radar data points are sparse, a radar data point is shared by some grid points in the vicinity of the radar data point. The radial shape in C- and X-SAPR plots is not as obvious because radar data points (VCP coverage in elevation angle space) are enough to avoid these behaviors. Because the density of radar data points decreases with distance from the radar for the NEXRAD radar (KVNX), which is also the furthest radar from the analysis domain, the radial shape problem becomes most evident in attempts to locate co-gridded regions for these datasets.

**Page 6 line 22: What is the time difference from the air density calculation and the analysis? For example, the air density within the squall line should be very different from the environment ahead of it. When are those values updated and the sensitivity to those changes applied.**

We agree with the reviewer that the air density can vary within MCSs and this is a potential source of uncertainty. As also mentioned in response to the reviewer's major comment #3, this study used air density profiles as from the ARM-provided Merged Sounding value-added product available at 1-minute intervals 266 altitude levels. The retrieval in this study used a profile nearest the analysis time from the product as the background. This assumption has been commonly used for multi-Doppler radar retrievals. As with other responses, we believed this analysis for the impact of air density was beyond the scope of this manuscript.

However, since the role of air density may not be obvious, again for the benefit of this response to our reviewers, we have attempted a simple sensitivity test to better demonstrate the potential impact of air density (profile variability) on 3DVAR wind retrievals. Three different air density profiles as shown in a figure below (Figure R2) were used for an interesting convective time around 1037 UTC on May 20, 2011: the control profile (black) and $\pm 2$ kg m$^{-3}$ of the control (blue and red) have been tested. The figure also shows 50, 75, 90, and 95 percentiles of retrieved updraft and downdraft values as one reference for potential impacts on key storm metrics. In the case of these larger profile shifts in air density, the differences between the three sets of retrievals are relatively small. In that sense, we believe that the impact of air density variability (e.g., perhaps to include spatial variability) was generally not significant for influencing wind field retrievals to the same level as other considerations we highlighted in this manuscript.

[Figure]

Figure R2: Vertical profiles of air density used for a sensitivity analysis (left) and probability density percentiles of updraft (middle) and downdraft(right).

**Page 8 line 18: Is there a citation or study that shows the velocity measurement error can be 0.5 m s⁻¹. Need to justify using 0.5 m s⁻¹ as the top of the error.**

Agree. The standard deviation of radial velocity ($V_{SD}$) depends on several factors including the sampling of the radars, SNR, and spectrum width, as discussed in several texts (e.g., Doviak and Zrnić 1993; Bring and Chandrasekar, 2001). We believe the 0.5 m s⁻¹ value that we employed is a useful value for sensitivity testing – one that may be consistent with a modest SNR value for severe weather event echoes (e.g., 20 dB), having an average of median spectrum width, 2.4 m s⁻¹ (Fang et al, 2004). The $V_{SD}$ could be larger than 0.5 m/s in these convective situations, e.g., reach ~1.0 m s⁻¹ at larger spectrum width values. Again, this $V_{SD}$ will also depend on sampling / scan speed (Bharadwaj, 2014, personal communication). $V_{SD}$ values are around 0.5 m s⁻¹ with echo SNR ~ 40 dB for an average scan speed used during MC3E (e.g., 18-20 degrees/sec). We have revised statements on these items in the updated manuscript.

**Page 9 Line 6: I would switch this paragraph with the one after it to help readability.**

We reorganized sections and figures. The paragraph and figure you pointed out were moved in section 5.

**Page 9 Line 28: Move the last sentence to the beginning of the sensitivity analysis to give the reader the ability to know where to reference as you describe.**

Done.

**Page 10 Line 18: How the new $R_s$ was determined to be 750 m?**

We empirically determined this value for Rs. We believe that this appears reasonable as it

represents approximately the horizontal distance of beam width (8 degrees ~ 1.5 km) of the RWPs, e.g. at around 10 km altitude.

**Page 11 Line 20-23: These two sentences seem out of order.**

We modified those sentences to read "The most prominent features in both retrievals are a deep updraft region above 3 km altitude and strong updraft values greater than 8 m s$^{-1}$."

**Page 11 Line 25: Move the detail about the time axis for the plots higher up to where the plot is initially discussed near line 20 so the change from the expected axis is noted earlier.**

Done.

**Page 12 Line 19: How do we know that this upwards motion is associated with the squall line convergence zone when there is not plot to give context that relate the two together.**

Some strong convergence and updraft were shown in Fig. 6. However, we decided to remove this sentence from the text.

**Chapter 6: How many times was the iterative upward technique iterated over the entire column? What was the method used to determine if/when the iterative technique converged on a solution? You mention number of iterations for the 3DVAR technique but none for the upward iterative technique.**

We iterated the solution of the iterative upward technique until the gradient of cost function falls 0.01. Generally, the iterative upward technique converged with about 220 iterations.

**Page 14 Line 8-9: This sentence should be removed or changed to not be as strongly worded as there has been other studies where multiband radars have been used together for analysis. See papers resulting from the VORTEX2 field project.**

We rewrote section 7 and decided to leave out the sentence from the text.

**Page 14 Line 27: It is hard for me to accept that the iterative wind retrieval is inferior to the 3DVAR technique from only 1 example and one that did not use advection correction.**

We modified this sentence to be more specific as "Overall, the 3DVAR technique can

produce smaller errors in updraft retrievals than the iterative upward integration technique particularly for severe convection events including large areas of strong convection." We extended the comparison analysis between the 3DVAR and the iterative upward integration technique to the five cases and discussed the advection effect in section 7.

---

## Author Comment (AC3) · 4 Jun 2017

**Response to Reviewer Comment #2**

We thank this reviewer very much for your positive comments on our manuscript. Providing this valuable feedback has helped to improve the current manuscript. Taking into account comments from all reviewers, we reorganized the manuscript sections as described to the previous reviewer and in our comments to this reviewer. The following contains our detailed responses to your comments, with our responses in plain text given underneath your original comments in bold type.

**1) In the conclusions, the authors assert "X-, C-, and S-band scanning radars have been used together to pseudo simultaneously": the reviewer is not able to find information about the S-band radar in Table 2 and its role in the study is not clear.**

Since we have rewritten section 7, this sentence has been removed from the text. However, the S-band radar in question is a NEXRAD radar (KVNX), for which we have assumed most readers are familiar with its general operation and modes. We have provided an additional reference. Its role in this study includes providing additional surveillance and unattenuated reflectivity coverage of convective cells. We believe the ability to better bound the convective storms from available NEXRAD insights improves these variational retrievals.

**2) There are no information about radiosonde location, please provide them updating also Figure 1.**

We used the ARM Merged Sounding dataset. This is a 'value added product' from ARM that uses a combination of observations from the radiosonde launched at the SGP CF (e.g., Lamont, Oklahoma), microwave radiometers, surface meteorological instruments, and European Centre for Medium Range Weather Forecasts (ECMWF) model output. The dataset is better described in section 2 of the revised manuscript.

**3) The observation simultaneity is the key factor for multi-Doppler retrievals: please provide more details on this topic.**

As in response to the previous reviewer #1 (and discussion image provided to that reviewer), this paper did not directly consider advection and time evolution of clouds. We believe this is a subject beyond the scope of this manuscript to address in the proper detail that may be required. As in our response to reviewer #1, we have added discussion on this topic to section 7.

**4) The comparison between 3DVAR and iterative retrievals is performed for only one event. It should extended to all cases**.

Agree. We extended the comparison analysis between the two techniques to the five MC3E

cases listed in Table 1. The RMSE of radial velocity and the normalized mass continuity residual (NMCR) estimated from the 3DVAR and an iterative upward integration method are listed in Table 4. Figure 6 shows an example of comparison of retrieved vertical velocity and horizontal divergence, and vertical profiles of RMSE and NMCR for the May 20 case. As shown in Table 4, the 3DVAR technique provides lower NMCR and radial velocity RMSE values than the upward integration technique for the five cases. However, the NMCR values from the April 25 and May 11 cases are low and relatively similar performance is found between the two techniques. Those two events are nocturnal elevated convection (April 25) and widespread stratiform precipitation with embedded convection (May 11), respectively. Both cases included narrow/weaker convective regions, and the propagation speeds for the cells were slower. This is in contrast to the advantages for the remaining convective cases that featured severe/significant convection and MCS systems having larger convective coverage. The result suggests to us that the iteration upward integration technique is still reasonably-matched for those two weaker convective cases where the mass continuity equation is a dominant parameter (aka, no significant advantage was gained), whereas for severe MCS convection cases, the 3DVAR technique is arguably preferable. We added this discussion to section 5, which has been moved from section 6 in the previous manuscript.

---

## Author Comment (AC4) · 4 Jun 2017

**Response to Reviewer Comment #3**

Thank you very much for your appreciation of our manuscript and giving valuable suggestions that have helped to improve the manuscript. Taking into account comments from all reviewers, we reorganized the sections. The following contains our detailed responses to your comments, with our responses in plain text given underneath your original comments in bold type.

**Specific comments:**
**1. Page 4, lines 1-7. The first line of page 4 states, "Data were calibrated and processed according to several standard methods in the literature, . . ." The manuscripts needs to clarify what ARM data are used in this study. Specifically, were the data calibrated and processed by ARM or by the authors of this manuscript. Also, the manuscript should include the DOI numbers of the ARM datasets used in this study and whether or not the data processed by this research team are available for others to analyze. Line 7 on page 4 states that the radar data are mapped to a common Cartesian analysis domain. I thought ARM already produced a radial moments to Cartesian coordinate moments data product. The manuscript needs to clarify whether or not this common Cartesian data is the same or different than an ARM data product.**

We have included additional information on the processing and datasets / availability. Radar reflectivity observed by CSAPR-I7 was corrected for attenuation in rain using the CSAPR-I7 specific differential phase (Kdp) measurements, as implemented using previously published methods. These include those as available also from ARM open-source python codes (Bringi and Chandrasekar, 2001; Giangrande et al., 2013b, 2014; Helmus and Collis, 2016). Because XSAPR reflectivity were significantly attenuated in rain (sometimes extinguished through heavier rain), for the XSAPRs, we only consider the mean Doppler velocity measurements in these retrievals (e.g., as mean Doppler velocity measurements are far less influenced by partial attenuation in rain). Aliased radial velocity measurements from all radars were corrected / dealiasied using the four-dimensional technique described in James and Houze (2001). We improved our description for this in section 2.1 of the revised manuscript.

In addition, we have added appropriate citations for the ARM datasets (Atmospheric Radiation Measurement (ARM) Climate Research Facility. 1996, 2011) to the text, and references including the DOI#s for those raw datasets. Nevertheless, processed radar datasets of this sort are generally unavailable to be placed immediately on the ARM archive. In part, this is because the accepted PI-products (processed) would typically not be released to that archive until those datasets are associated with a formal publication.

For this study, we mapped all scanning radar datasets (CSAPR, 3 XSAPRs, and NEXRAD KVNX radar) to the common Cartesian domain with 0.25-km horizontal and vertical grid spacings. This is different from the ARM Mapped Moments to a Cartesian Grid (MMCG) Value Added Product, which has a 240 km x 240 km domain with 1-km horizontal and 0.5-km vertical spacings. We described the domain size and spatial resolution settings and the interpolation method used in this study in section 2. Since this particular ARM product

is not used (e.g., we start from the raw ARM datastreams), we did not see need to describe the differences between our gridding and those from the ARM Cartesian coordinate data products. However, we can understand if there is some confusion. Moreover, we need to differentiate these efforts since those MMCG products would not have the same detail/attention for velocity dealiasing or attenuation correction in rain as what was done for this study (both corrections must happen in radial coordinates prior to gridding). We do anticipate that future ARM radar products (CMAC-series) should have appropriate corrections for attenuation in rain and velocity dealiasing prior to gridding. Some of these features have already been made publically available for some transparent processing through ARM's open-source python radar processing toolkits, Py-ART. What those codes cannot replicate is any manual (investigator) quality control checks, filtering or similar that semi-automatic processing (no matter how good) can likely replicate (a subject that is still a problem for storm scale velocity dealiasing in particular for this community).

**2. Page 4, lines 15 to 19. The manuscript should state the weights used in this study. As written, the manuscript describes that the weights are important, but not the actual weight values. If this work is to be repeatable by others, then the weights of the gridding should be published in this manuscript.**

Weights calculated by Eq (1) were used to map the radar data to the Cartesian coordinate domain. Figure 3 in the revised manuscript shows nearest neighbor distance at each gird point and its weights. We also used the nearest neighbor weights as the observation constraint weights of the cost function in the 3DVAR retrieval (section 3.1).

**3. Page 4, lines 1 to 19. The manuscript needs to describe in this section what radar observations are used in this manuscript. With the importance of preserving the phase and amplitude information of the input radar data (see line 19), the reader is led to believe that a phase measurement (e.g., Kdp) is used in this study. But in later pages, it appears that only reflectivity and radial velocity are used in this study.**

As in previous responses and our revised manuscript, we used 4 radars from the ARM scanning precipitation radar network and one radar from the NEXRAD WSR-88D S-band radar network. The ARM radar network includes a 6.3-GHz C-band scanning ARM precipitation radar (CSAPR-I7) and three 9.4-GHz X-band scanning ARM precipitation radars (XSAPRs, named I4, I5, and I6) for the multi-Doppler radar wind retrieval.

As noted, in the revised manuscript, radar reflectivity values as observed by CSAPR-I7 are significantly attenuated in rain and are needing to be corrected for attenuation in rain using the CSAPR-I7 differential phase measurements. We have removed mention of 'KDP' to avoid additional confusion. While basic procedures are part of dual-pol XSAPR reflectivity processing, these measurements were more significantly attenuated in rain, thus we only rely on the mean Doppler velocity measurements from the XSAPR for these retrievals (e.g., ultimately XSAPR correction is a nonfactor in our results). Similarly, NEXRAD radar observations are from the dual-polarization NEXRAD (KVNX was upgraded prior to

MC3E in 2011) may also be corrected for attenuation using differential phase measurements. In an equal, but opposite manner, it should be noted that performing the correction methodologies for attenuation in rain at S-band are arguably also less important to our findings (e.g., S-band being relatively unattenuated wavelength). Aliased radial velocity measurements from all radars were corrected using the four-dimensional technique described in James and Houze (2001). We described this in section 2 in the revised manuscript.

The ARM zenith-pointing radar wind profilers at two locations, UAZR-C1 and UAZR-I9, were used to evaluate the multi-Doppler wind retrieval. These locations are shown in Fig. 1.

**4. Page 5, line 14. This reviewer noticed the phrase "surface impermeability". That is a fancy way of saying "surface boundary condition".**

"Surface impermeability" was introduced by Scialom and Lemaître (1990) as a vertical velocity boundary condition at the ground level. This concept has been used in the 3DVAR analysis by several papers, and this phrase has been conventionally used (e.g., Shapiro and Mewes 1999). Therefore we decided to use this phrase in this paper and added the following sentences to section 3.3: "This study imposes surface impermeability (Scialom and Lemaître 1990) as a vertical velocity boundary condition at the ground level.".

**5. Page 5, lines 13-22. I found this section hard to read because the cost function terms are not defined. The manuscript needs to define the cost functions of Jo, Jc, Jp, Jb, and Js. As written, these terms are not introduced until subsequent section headings.**

Thank you for pointing out. We defined the physical constraints of radial velocity observations ($J_o$), anelastic mass continuity ($J_c$), surface impermeability ($J_p$), background wind field, ($J_b$), and spatial smoothness ($J_s$) in the beginning of section 3.

**6. Page 8, line 25 and onwards. The variable names for the weights are different in the text and in the figures. These different variable names is very confusing for the reader and the manuscript needs to be corrected.**

See our response to comment # 5.

**7. Table 2. I found the entries for pulse width and range resolution to be confusing and possibly redundant. The entry for the UAZR range resolution is either 200 meters or 120 meters, not both, please correct or clarify.**

The radar wind profilers operate using two alternating modes, a long pulse mode (200 m gate spacing, to ~15 km, ~20 m/s Nyquist) and a short pulse mode (120 m, to ~9 km, ~14 m/s Nyquist). The reviewer is correct that all that really matters is that we have a single

merged profiler dataset (at the 200 m resolution) that combines those two modes (a staggered PRT approach also assists in dealiasing of profiler velocities), e.g., as in Tridon et al. [2013], etc. We added parameter settings of the two modes to Table 2.

---

## Author Comment (AC5) · 4 Jun 2017

**Response to Reviewer Comment #4**

Thank you very much for your valuable suggestions that have helped to improve the manuscript. Taking into account comments from all reviewers, we reorganized the sections. The following contains our detailed responses to your comments, with our responses in plain text given underneath your original comments in bold type.

**The paper has a thorough introduction explaining the importance and current understanding and well presented graphics. The paper answers many questions I initially would ask – e.g. the attenuation issues and need for S-band in these cases. I appreciated the descriptions of the cases in section 5.2. However, I feel there are some "holes" that need to be filled for publication.**

We hope we can address some of the reviewer comments in our responses below.

**- Page 4 – the calibration and processing is essentially stated as standard, yet these practices are certainly far from universal and require more detail on what calibration, processing and quality control has occurred.**

This question has come up with other reviewers. We have revised section 2 and added an improved description of radar data processing. Of most importance to this study, aliased radial velocity measurements from all radars were corrected using the four-dimensional technique described in James and Houze (2001). Similar to Collis et al. (2013), this technique was applied iteratively using multiple wind profiles from the MC3E radiosonde network to produce robust results (e.g., Jensen et al., 2015b). Finally, each radar volume was manually inspected to check for conspicuous errors and artifacts.

**- Equation 1 requires weights, which are subsequently mentioned, but the chosen values are never given – for repeatability they are important.**

Weights calculated by Eq. (1) are used to map the radar data to the Cartesian coordinate domain. The weights decrease with distance between the grid point and a radar data point. Figure 3 in the revised manuscript shows nearest neighbor distance at each gird point and its weights. We also used the nearest neighbor weights as the observation constraint weights of the cost function in the 3DVAR retrieval (section 3.1).

**- I found section 3 hard to follow, some symbols seem unspecified (at least until later) or inconsistent (notably don't match the plots) which is I believe the source of my initial difficulty.**

Thank you for pointing these issues out to the authors. We defined the physical constraints of radial velocity observations ($J_o$), anelastic mass continuity ($J_c$), surface impermeability ($J_p$), background wind field, ($J_b$), and spatial smoothness ($J_s$) in the beginning of section 3. We also revised descriptions about $\Lambda$. In the manuscript, capital lambda ($\Lambda$) represents a $n \times n$ matrix of constraint weights where $n$ is identical to the number of analysis points,

while lower case lambda represents diagonal element of the matrix $\boldsymbol{\Lambda}$, which is treated as adjustable parameters. Although the matrix $\boldsymbol{\Lambda}$ has a number of elements, this study uses a constant values of $\lambda$ for $\boldsymbol{\Lambda}_c$, $\boldsymbol{\Lambda}_p$, $\boldsymbol{\Lambda}_b$, and $\boldsymbol{\Lambda}_s$. We specified this in section 3 and revised text, tables, and figures appropriately.

**- the beginning of section 7 suggests "X-, C-, and S-band scanning radars have been used together to pseudo simultaneously", yet the S-band seems only used for reflectivity in the case where attenuation is thought to be significant, and it isn't clear if it is used at all in other cases.**

Reflectivity and Doppler velocity measurements from a NEXRAD WSR-88D S-band radar (KVNX) are used for all retrieval cases.
We rewrote  section 7 and have decided to leave out the sentence from the text.

**- I'd like to see the S-band radar in table 2 if it is intended to be used together as in the previous point.**

Done.